# Modulating Effect of Carbohydrate Antigen 125 on the Prognostic Value of High-Sensitivity C-Reactive Protein in Heart Failure

**DOI:** 10.3390/biom15091260

**Published:** 2025-08-30

**Authors:** Enrique Santas, Arancha Martí-Martínez, Elena Revuelta-López, Sandra Villar, Rafael de la Espriella, Patricia Palau, Pau Llàcer, Gema Miñana, Enrique Rodriguez-Borja, Arturo Carratalá, Arantxa Gonzalez, Antoni Bayés-Genís, Juan Sanchis, Julio Núñez

**Affiliations:** 1Cardiology Department, Hospital Clínico Universitario de Valencia, Instituto de Investigación Sanitaria (INCLIVA), 46010 Valencia, Spain; ensantas@gmail.com (E.S.); sanvc28@gmail.com (S.V.); delaespriella_raf@gva.es (R.d.l.E.); patripalau@gmail.com (P.P.); gemineta@gmail.com (G.M.); sanchis_juafor@gva.es (J.S.); 2Faculty of Medicine and Dentistry, Universitat de València, 46010 Valencia, Spain; 3Clinical Biochemistry Department, Hospital Clinico Universitario de Valencia, 46010 Valencia, Spain; aranchamartimartinez@gmail.com (A.M.-M.); rodriguez_enr@gva.es (E.R.-B.); arturocarratalacalvo@gmail.com (A.C.); 4Centro de Investigación Biomédica en Red Enfermedades Cardiovasculares (CIBERCV), 28029 Madrid, Spain; erevuelta@igtp.cat (E.R.-L.); amiqueo@unav.es (A.G.); abayesgenis@gmail.com (A.B.-G.); 5Cardiology Department, Hospital Universitari Germans Trias i Pujol, Universitat Autònoma de Barcelona, 08916 Badalona, Spain; 6Internal Medicine Department, Hospital Universitario Ramón y Cajal, 28029 Madrid, Spain; paullacer@hotmail.com; 7Program of Cardiovascular Disease, CIMA Universidad de Navarra, Department of Cardiology and Cardiac Surgery, Clínica Universidad de Navarra and IdiSNA, 31008 Pamplona, Spain

**Keywords:** acute heart failure, CA125, hs-CRP, inflammatory modulator

## Abstract

Inflammation and congestion are key pathophysiological processes in heart failure (HF). Our aim was to evaluate the potential modulatory effect of carbohydrate antigen 125 (CA125) on inflammation, assessed by high-sensitivity C-reactive protein (hs-CRP). We analyzed a cohort of 4043 consecutive patients in whom hs-CRP and CA125 levels were measured during a hospitalization for acute HF. Multivariate Cox regression models were applied to assess the association between the biomarkers and all-cause mortality and death/HF rehospitalization at 6 months. In multivariable analysis, a significant interaction between hs-CRP and CA125 was observed for both outcomes (*p*-value for interaction = 0.036 and <0.001, respectively). hs-CRP was significantly associated with an increased risk of death (HR = 1.27; 95% CI 1.16–1.41; *p* < 0.001) and death/HF rehospitalization (HR = 1.18; 95% CI 1.09–1.28; *p* < 0.001) if CA125 > 35 U/mL. In contrast, hs-CRP was not predictive of events when CA125 ≤ 35 U/mL. In conclusion, in patients with acute HF, the association between hs-CRP and clinical outcomes was modulated by CA125 levels. hs-CRP was associated with a higher risk of events only in patients with elevated CA125. These findings support a potential modulatory and amplifying role for CA125 in the inflammatory response in HF.

## 1. Introduction

Heart failure (HF) is a complex clinical syndrome involving multiple pathophysiological mechanisms that contribute to substantial morbidity and mortality worldwide [1]. Among these, inflammation and congestion are closely intertwined and play a pivotal role in HF progression [2,3,4].

Systemic inflammation contributes to HF progression by promoting adverse cardiac remodeling, endothelial dysfunction, oxidative stress, and fluid overload [2,5,6]. In particular, activation of the interleukin-1 (IL-1)/interleukin-6 (IL-6) signaling axis is strongly associated with disease severity and prognosis [2,5,6,7,8]. High-sensitivity C-reactive protein (hs-CRP), an acute-phase reactant synthesized in response to NLRP3 inflammasome activation, is widely used in clinical practice as a surrogate marker of systemic inflammation [9,10]. Accordingly, elevated hs-CRP levels have been associated with an increased risk of morbidity and mortality in patients with HF [11,12]. On the other hand, circulating levels of carbohydrate antigen 125 (CA125) have emerged as a useful biomarker in HF, with robust validation due to its strong association with congestion, particularly volume overload and interstitial fluid accumulation, and prognosis [13,14,15,16]. Beyond its association with congestion, CA125 expression is upregulated by inflammatory stimuli such as TNF-α, IL-1β, or IL-6, suggesting a potential interaction in inflammatory pathways in HF [13,14]. Structurally, CA125 possesses a highly glycosylated extracellular domain, which enables interaction with immune-related proteins [17]. Recent evidence suggests that CA125 may act not only as a biomarker, but also as a functional ligand capable of modulating inflammatory responses [18,19].

In acute HF, previous studies have shown that the prognostic relevance of inflammation and fibrosis-related biomarkers, such as galectin-3 (Gal-3) and soluble ST2 (sST2), is amplified in the presence of elevated CA125 levels [20,21]. These findings support the hypothesis that CA125 may modulate inflammatory pathways and influence the prognostic value of other biomarkers.

Therefore, our study aimed to evaluate whether CA125 modifies the prognostic influence of hs-CRP in patients with HF by assessing its interaction with hs-CRP in predicting adverse clinical outcomes in patients with HF.

## 2. Materials and Methods

### 2.1. Study Group and Protocol

This is a retrospective analysis of a prospective registry including 5222 consecutive patients admitted from January 2012 to December 2021 for acute HF in three hospitals in Valencia, Spain. We excluded patients without assessment of hs-CRP during admission (*n* = 1179), leaving the final study sample of 4043 patients. A comprehensive dataset was collected during the index hospitalization using pre-established registry questionnaires, including demographic characteristics, medical history, laboratory values, echocardiographic parameters, and treatments at discharge. Patients with either new-onset or worsening HF were eligible for inclusion. Acute HF was defined according to the contemporary European Society of Cardiology Clinical Practice Guidelines applicable throughout the study period. Treatment strategies were individualized based on the attending physicians’ clinical judgment and in accordance with guideline-directed management at the time of inclusion.

The study conformed to the principles outlined in the 1975 Declaration of Helsinki and was approved by the institutional ethics committee. All patients provided written informed consent.

### 2.2. Biomarkers Measurement

Plasma levels of hs-CRP and CA125 were routinely obtained within the first 24 h after admission. Both were measured using standard commercial enzyme immunoassays: hs-CRP by turbidimetric immunoassay Alinity-c CRP Vario Reagent Kit (Abbott^®^ Laboratories, Abbott Park, IL, USA) and CA125 by microparticle chemiluminescent immunoassay Roche Elecsys^®^ CA 125 II (Roche Diagnostics, International, Rotkreuz, Switzerland).

### 2.3. Follow-Up and Endpoints

All-cause mortality at 6 months and the combined endpoint of all-cause mortality or unplanned HF rehospitalizations within 6 months were selected as the primary endpoints. Only unplanned hospital readmissions were considered. Outcome assessment was conducted by verifying patient survival status and hospital readmissions through a systematic review of electronic medical records from the public healthcare system of the Valencian Community. This process utilized data from the SIA-GAIA and Orion Clinics electronic databases, comprehensively capturing all medical encounters within the regional public health system. Clinical events were verified through review of electronic medical records and adjudicated by an investigator who was blinded to clinical information and biomarker levels.

### 2.4. Statistical Analysis

Continuous variables are presented as mean ± standard deviation (SD) or median (interquartile range [IQR]: p25 to p75), as appropriate. Categorical variables are expressed as frequencies (percentages). Given the absence of a universally accepted hs-CRP threshold in acute HF, hs-CRP was analyzed both as a continuous variable and dichotomized by the median: 14.9 mg/L (IQR: 6–36.2). CA125 was dichotomized using the established cut-off of 35 U/mL. A composite categorical variable in including both biomarkers was generated by combining both hs-CRP and CA125 thresholds: C1 = hs-CRP ≤ 14.9 mg/L and CA125 ≤ 35 U/mL (*n* = 795); C2 = hs-CRP > 14.9 mg/L and CA125 ≤ 35 U/mL (*n* = 786); C3 = hs-CRP ≤ 14.9 mg/L and CA125 > 35 U/mL (*n* = 1236), and C4 = hs-CRP > 14.9 mg/L and CA125 < 35 U/mL (*n* = 1236).

Group comparisons were performed using the χ^2^ test for categorical variables and ANOVA or the Kruskal–Wallis test for continuous variables, depending on data distribution. To evaluate the modulating effect of CA125 on the association between hs-CRP and clinical outcomes, we stratified the analysis by CA125 levels (≤35 and >35 U/mL). Associations were assessed using Cox proportional hazard regression, and *p*-values for interactions were calculated. Risk estimates were expressed as hazard ratios (HR) with 95% confidence interval (95% CI).

All variables listed in Table 1 were tested for multivariable adjustment based on prior knowledge/biological plausibility, regardless of their univariate *p*-value. The linearity assumption for continuous variables was tested, and transformations using fractional polynomials were applied as needed. A parsimonious model was subsequently derived using backward stepwise selection guided by clinical judgment. The covariates included in the multivariable model for 6-months mortality were: age, sex, first HF admission, last known under stable conditions New York Heart Association (NYHA) class, atrial fibrillation, left bundle branch block, peripheral edema, pleural effusion, systolic blood pressure at admission, heart rate at admission, estimated glomerular filtration rate, hemoglobin, sodium, left ventricular ejection fraction (LVEF), tricuspid annular systolic plane excursion (TAPSE) by echocardiography, left atrial diameter, clinical diagnosis of acute infection, and discharge treatments (diuretics, beta-blockers, mineralocorticoid receptor antagonists, renin-angiotensin-aldosterone system (RAAS) inhibitors, and statins). The model for the composite 6-month endpoint included the following covariates: age, sex, first HF admission, last known under stable conditions NYHA class, atrial fibrillation, left bundle branch block, peripheral edema, pleural effusion, systolic blood pressure at admission, heart rate at admission, estimated glomerular filtration rate, LVEF, TAPSE, left atrial diameter, and discharge treatments (diuretics, beta-blockers, mineralocorticoid receptor antagonists, and RAAS inhibitors).

A two-sided *p*-value of <0.05 was considered statistically significant for all analyses. All survival analyses were performed using STATA 18 (StataCorp. LLC, College Station, TX, USA).

## 3. Results

The median (IQR) age of the cohort was 77 years (68–83), and 1833 patients (45.3%) were women. The mean LVEF was 49.5% ± 15.3 and 2174 (53.8%), and 1869 (46.2%) displayed LVEF < 50% and ≥50%, respectively. The median hs-CRP and CA125 levels were 14.9 mg/L (IQR: 6.0–36.2) and 50.2 U/mL (IQR: 22.2–114.5). Baseline characteristics of the cohort are summarized in Table 1 and Table A1. Notably, there were no significant differences in hs-CRP levels across CA125 strata (≤35 or >35 U/mL; *p* = 0.223).

### 3.1. Baseline Characteristics Across hs-CRP/CA125 Categories

Patients with both biomarkers elevated (hs-CRP > 14.9 mg/L and CA125 > 35 U/mL) exhibited a worse baseline clinical profile. This group included a higher proportion of men and showed a greater prevalence of atrial fibrillation, and presented with lower blood pressure at admission. Clinical signs of systemic congestion, such as pleural effusion and peripheral edema, were also more prevalent in this group. Laboratory findings revealed a higher prevalence of renal dysfunction, lower hemoglobin levels, hyponatremia, and elevated NT-proBNP levels across biomarker categories, with a stepwise increase from category 1 (both biomarkers below the cut-offs) to category 4 (both elevated). Regarding treatment, patients in category 4 were more likely to receive more intensive diuretic regimens, but less frequently received guideline-directed medical therapy (Table 2 and Table A2).

### 3.2. hs-CRP and 6-Month Adverse Outcomes

At the 6-month follow-up, 538 patients (13.3%) had died, and 1106 (27.4%) experienced the composite outcome of death or unplanned HF readmission.

In the overall cohort, patients with hs-CRP levels above the median exhibited significantly higher rates of all-cause mortality (15.9% vs. 10.7%, *p* < 0.001) and the composite outcome of 6-month death or HF readmission (30.6% vs. 24.2%, *p* < 0.001). These associations remained significant after multivariable adjustment. Specifically, hs-CRP > 14.9 mg/L was independently associated with an increased risk of mortality (HR: 1.22; 95% CI: 1.02–1.46; *p* = 0.031) and the composite endpoint (HR: 1.36; 95% CI: 1.11–1.66; *p* = 0.003).

### 3.3. hs-CRP and Clinical Outcomes Across CA125

We found a stepwise increase in the incidence of both endpoints across the four biomarker-defined categories, from C1 (hs-CRP < 14.9 mg/L and CA125 < 35 U/mL) to C4 (hs-CRP ≥ 14.9 mg/L and CA125 ≥ 35 U/mL). Six-month mortality rates were 8.3% in C1, 10.2% in C2, 12.3% in C3, and 19.5% in C4 (*p* for trend < 0.001). Corresponding rates for the composite endpoint were 20.3%, 25.4%, 26.6%, and 33.9%, respectively (*p* for trend < 0.001). These differences emerged early during follow-up and progressively widened over time, as confirmed by Kaplan–Meier survival analyses for both all-cause mortality (Figure 1A) and the composite outcome (Figure 1B).

Following multivariable adjustment, we confirmed a significant heterogeneous association between hs-CRP and adverse clinical events across CA125 levels (interaction *p* values = 0.036 and <0.001 for mortality and the composite endpoint, respectively). For mortality, hs-CRP was not associated with risk among patients with CA125 ≤ 35 U/mL (Figure 2a, *p* = 0.828). In contrast, among those with CA125 > 35 U/mL, hs-CRP displayed a significant and linear association with increased mortality risk (Figure 2b, *p* < 0.001). Similar results were found when hs-CRP was analyzed as a binary variable: in the low CA125 group, hs-CRP above the median was not linked to increased mortality (HR: 0.93; 95% CI: 0.49–1.75; *p* = 0.828), whereas in the high CA125 group, it was associated with greater risk (HR: 1.37; 95% CI: 1.22–1.55; *p* < 0.001). A comparable pattern was found for the composite outcome. Elevated hs-CRP was not significantly associated with the risk of death or HF readmission in patients with CA125 ≤ 35 U/mL (Figure 3a, *p* = 0.177), but was positively associated with increased risk in those with CA125 > 35 U/mL (Figure 3b, *p* < 0.001). Consistently, hs-CRP above the median independently predicted the composite endpoint in the high CA125 group (HR: 1.29; 95% CI: 1.18–1.39; *p* < 0.001), but not in the low CA125 group (HR: 1.13; 95% CI: 0.74–1.78; *p* = 0.505).

Risk estimates for all covariates included in the multivariable models are provided in Table A3 and Table A4.

## 4. Discussion

In patients with acute HF, we found that the prognostic relevance of systemic inflammation, as measured by hs-CRP, was significantly influenced by circulating CA125 levels. Specifically, elevated hs-CRP values were independently associated with an increased risk of all-cause mortality and HF readmissions only in patients with high CA125 concentrations. In contrast, hs-CRP was not significantly associated with adverse outcomes in patients with CA125 ≤ 35 U/mL. These findings align with prior observations indicating that the prognostic utility of other inflammation- and fibrosis-related biomarkers—such as sST2 and Gal-3—also depends on CA125 status [20,21].

Together, these findings reinforce the hypothesis that CA125 is not merely a passive marker of congestion but may functionally modulate inflammatory signaling cascades, thereby shaping the prognostic landscape in HF. This interaction highlights the convergence of two critical pathophysiological axes in HF—congestion and inflammation—and underscores the clinical relevance of their interplay.

### 4.1. Inflammation and Congestion: Two Converging Pathophysiological Axes in HF

Inflammation is a key pathological process in heart failure (HF) [2,5]. Through a complex network of cytokines and cellular mediators, inflammation exerts direct effects on both systolic and diastolic function, promoting interstitial collagen deposition, oxidative stress, and endothelial dysfunction, thereby contributing to myocardial dysfunction and multi-organ involvement [2,5]. Notably, a positive correlation between volume overload and systemic inflammatory markers, including hs-CRP or IL-6, has been observed in different studies [13,14,22].

Pro-inflammatory cytokines such as IL-6 and TNF-α increase vascular permeability by disrupting the endothelial barrier, facilitating fluid extravasation into the interstitial space, contributing to tissue congestion, and aggravating volume overload [3,22]. In parallel, inflammatory activation stimulates both the RAAS and the sympathetic nervous system, further promoting sodium and water retention, vasoconstriction, and hemodynamic overload. Conversely, congestion itself may act as a potent pro-inflammatory stimulus. Experimental models have shown that venous congestion can activate maladaptive inflammatory pathways [3,23]. Clinically, elevated central venous pressure compromises organ perfusion and contributes to congestion-induced organ injury [24,25]. For instance, intestinal congestion disrupts mucosal integrity, enabling the translocation of bacterial endotoxins into the systemic circulation and promoting systemic inflammation in HF [26].

Consequently, inflammation and congestion engage in a bidirectional feedback loop, whereby inflammation amplifies vascular permeability and fluid retention, thereby intensifying congestion, while congestion itself triggers inflammatory signaling pathways, further promoting fluid accumulation and extravascular volume expansion.

Importantly, this biological interplay may lead to a more malignant phenotype of volume overload, characterized by extravascular volume overload and tissue inflammation, which is not fully captured by traditional clinical or hemodynamic assessment [22,27]. Given the complex interplay between inflammation and congestion, investigating the role of biomarkers—particularly the potential interaction between hs-CRP and CA125—may yield valuable insights into their prognostic relevance in HF.

### 4.2. CA125 as a Biomarker Beyond Congestion, a Modulator of Inflammatory Pathways?

CA125, also known as mucin 16 (MUC16), is an established biomarker in HF. The activation of serosal cells due to volume overload and inflammation is believed to be the key mechanism driving the upregulation of this biomarker [13]. Clinically, it has robust prognostic value, primarily due to its strong correlation with signs of volume overload and extravascular congestion [13,14,15,16]. However, recent evidence suggests that CA125 may have functional roles beyond congestion. MUC16 is a large transmembrane mucin with several potential cleavage sites [28]. Cleavage of MUC16 results in the release of its N-terminal domain into the circulation, while the C-terminal domain (CTD) remains on the cell surface or may translocate to the nucleus, where it can act as a transcriptional co-regulator [29]. In malignancies such as ovarian and pancreatic cancer, CTD translocation activates the expression of invasion-related genes, contributing to disease progression. Additionally, the glycosylated extracellular domain of MUC16 facilitates protein-protein interactions that initiate intracellular signaling cascades.

Recent studies have identified CA125 CTD as a component of an N-glycan-mediated protein-binding complex that includes EGFR, β1 integrin, and Gal-3 on the cell surface [30,31]. These interactions promote epithelial-to-mesenchymal transition (EMT), a process implicated in cancer metastasis, organ fibrosis, and tissue remodeling. These mechanisms suggest that MUC16/CA125 may also influence inflammatory signaling and tissue remodeling in HF. Notably, MUC16 expression has been recently detected in epicardial adipose tissue in patients with HF, where it correlates with pro-fibrotic and inflammatory markers, including IL-6 and Gal-3 [19].

In this line, a potential modulatory effect of CA125 on inflammatory pathways has been postulated. In a previous study, the association between Gal-3 and adverse clinical outcomes was dependent on CA125 levels: higher Gal-3 levels were deleterious only in patients with elevated CA125 values, but not in others [20]. A similar interaction has been recently reported with sST2, a biomarker primarily linked to inflammatory and fibrotic pathways in HF. In a subanalysis of the IMPROVE-HF trial, elevated sST2 levels predicted cardiovascular-renal rehospitalizations in patients with high CA125 levels (> 35 U/mL), but not in those with CA125 ≤ 35 U/mL [21]. These interactions suggest that CA125 may modulate the biological activity and prognostic significance of inflammatory pathways in HF.

An alternative explanation may relate to the relatively long half-life of CA125, which ranges between 7 and 12 days [13]. In this context, elevated hs-CRP levels in patients with concomitantly high CA125 may identify individuals with a heightened and more chronic inflammatory burden. Conversely, elevated hs-CRP in the presence of normal CA125 levels likely reflects an acute, transient inflammatory state rather than sustained inflammatory activity.

### 4.3. Potential Clinical Implications

From a clinical perspective, the coexistence of elevated hs-CRP and high CA125 identifies a ‘congestive-inflammatory’ phenotype, associated with increased worsening HF risk. In such cases, a comprehensive, multiparametric assessment of congestion and a more individualized depletive treatment strategy may be warranted [27]. Moreover, therapies aimed at attenuating systemic inflammation, reducing vascular permeability, and limiting tissue volume expansion could be particularly beneficial [32]. Future research should investigate whether targeted anti-inflammatory interventions confer prognostic or therapeutic advantages specifically in patients with elevated CA125 levels. In contrast, elevated hs-CRP in the context of low CA125 likely reflects a less concerning, transient inflammatory state, potentially requiring less intensive management. Accordingly, CA125 may function as a gatekeeper biomarker, refining the interpretation of inflammation-related risk in HF.

### 4.4. Limitations

Several limitations of this study should be acknowledged. First, the retrospective observational design is inherently subject to potential biases, including selection and information bias. Second, the observed biological interaction between CA125 and hs-CRP is exploratory and should be interpreted as hypothesis-generating. Further mechanistic and translational studies are required to elucidate the underlying pathways and validate the proposed modulatory role of CA125. Third, despite comprehensive adjustment for relevant covariates, the possibility of residual confounding cannot be excluded. Fourth, although the conventional 35 U/mL threshold for CA125 is widely adopted in oncology and heart failure research, it still lacks cross-platform harmonization and population-specific validation. Lastly, biomarker levels were assessed only at baseline; longitudinal measurements could have provided valuable insights into their temporal dynamics and evolving interplay during follow-up.

## 5. Conclusions

In patients with acute HF, the association between hs-CRP and 6-month adverse clinical events was modulated by CA125 status. hs-CRP was associated with a higher risk of events only in patients with elevated CA125 (>35 U/mL). Mechanistic translational studies are warranted to confirm these findings and unravel the biological mechanisms underlying them.

## Figures and Tables

**Figure 1 biomolecules-15-01260-f001:**
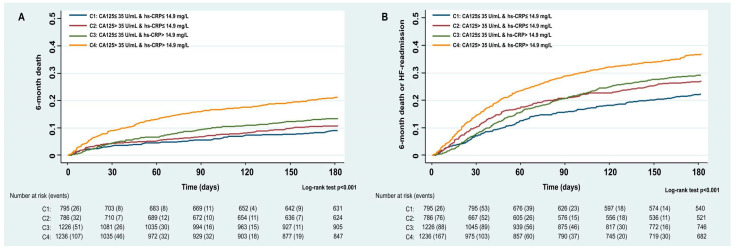
Kaplan–Meier curves between hs-CRP and CA125 categories for. (**A**) Six-month all-cause mortality; (**B**) six-month all-cause mortality and/or unplanned HF rehospitalization. CA125: Carbohydrate antigen 125; hs-CRP: High-sensitivity C-reactive protein.

**Figure 2 biomolecules-15-01260-f002:**
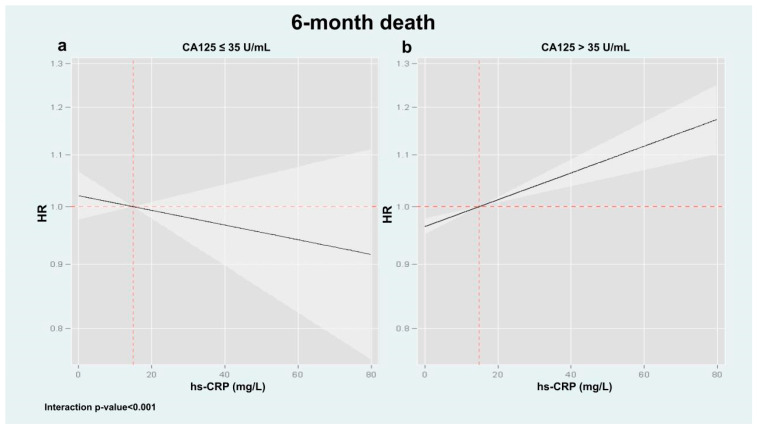
Association of hs-CRP with 6-month all-cause mortality stratified by CA125 status in multivariable models. A significant interaction was observed between hs-CRP and CA125 strata (interaction *p* < 0.001 for both outcomes). When CA125 was elevated (>35 U/mL), hs-CRP was linearly associated with an increased risk of all-cause mortality (panel (**b**)). In contrast, such an association was not found in patients with CA125 ≤ 35 U/mL (panel (**a**)). CA125: Carbohydrate antigen 125; HF: Heart failure; hs-CRP: High-sensitivity C-reactive protein.

**Figure 3 biomolecules-15-01260-f003:**
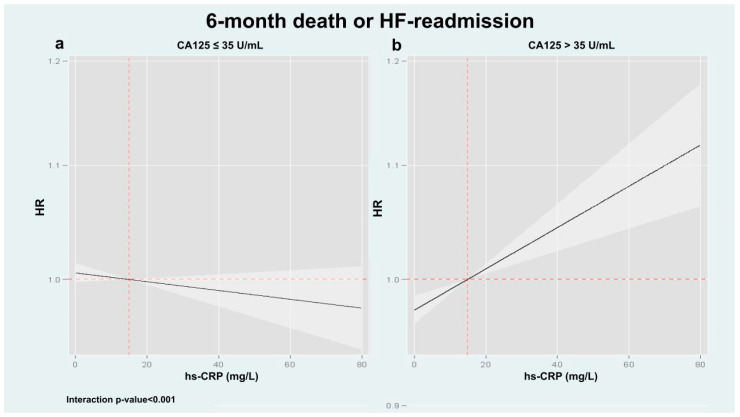
Association of hs-CRP with the composite 6-month endpoint (all-cause mortality and/or HF rehospitalization) stratified by CA125 status in multivariable models. A significant interaction was observed between hs-CRP and CA125 strata (interaction *p* < 0.001). When CA125 was elevated (>35 U/mL), hs-CRP was linearly associated with an increased risk of events (panel (**b**)). In contrast, hs-CRP was not associated with a higher risk of events if CA125 ≤ 35 U/mL (panel (**a**)). CA125: Carbohydrate antigen 125; HF: Heart failure; hs-CRP: High-sensitivity C-reactive protein.

**Table 1 biomolecules-15-01260-t001:** Baseline characteristics.

	Total Cohort (*n* = 4043)
Age, years ^a^	77 (68–83)
Female, *n* (%)	1833 (45.3%)
**Medical history**	
Hypertension, *n* (%)	3207 (79.3)
Diabetes mellitus, *n* (%)	1769 (43.8)
Dyslipidemia, *n* (%)	2175 (53.8)
HFpEF, *n* (%)	748 (57.0)
eGFR (MDRD formula), mL/min/1.73 m^2^	62.5 ± 28.2
Ischemic heart disease, *n* (%)	1298 (32.1)
First hospital admission, *n* (%)	2749 (68.0)
NYHA III-IV, *n* (%)	662 (16.4)
Atrial fibrillation, *n* (%)	1864 (46.1)
**Vital signs**	
Heart rate, bpm	95.9 ± 27.5
SBP, mmHg	143.4 ± 31.0
DBP, mmHg	79.9 ±19.0
**Echocardiography**	
TAPSE	18.6 ± 3.7
Septum, mm	11.9 ± 2.7
DTDVI, mm	54.4 ± 9.3
LVPW, mm	11.3 ± 2.0
**Laboratory**	
Hemoglobin, g/dL ^a^	12.5 (11.1–13.9)
Creatinine, mg/dL ^a^	1.1 (0.9–1.5)
Sodium, mEq/L	138.3 ± 4.5
NT-proBNP, pg/mL ^a^	3723.0 (1977.8–7694.0)
hs-CRP, mg/L ^a^	14.9 (6.0–36.2)
CA125, U/mL ^a^	50.2 (22.2–114.5)
**Treatment**	
ACE-I/ARB/ARNI, *n* (%)	2423 (59.9)
MRA, *n* (%)	1303 (32.2)
Diuretics prescribed at discharge, *n* (%)	3754 (93.6)
Beta-blockers prescribed at discharge, *n* (%)	2816 (69.7)
Statins prescribed at discharge, *n* (%)	2030 (50.2)

^a^ Value expressed as mean ± standard deviation, medians (percentile 25–percentile 75), or numbers (percentages). ACE-I: Angiotensin converting enzyme inhibitors; ARB: Angiotensin receptor blocker; ARNI: Angiotensin receptor–neprilysin inhibitor; CA125: Antigen carbohydrate 125; DBP: Diastolic blood pressure; DTDVI: Left ventricle telediastolic diameter; eGFR: Epidermal growth factor receptor; hs-CRP: High-sensitivity C-reactive protein; HFpEF: Heart failure with preserved ejection fraction; LVPW: Left ventricular posterior wall; MRA: Aldosterone receptor antagonists; NT-proBNP: N-terminal pro-B-type natriuretic peptide; NYHA: New York Heart Association; SBP: Systolic blood pressure; TAPSE: Tricuspid annular plane systolic excursion.

**Table 2 biomolecules-15-01260-t002:** Baseline characteristics across hs-CRP and CA125 categories.

	Category 1 Low hs-CRP and Low CA125 (*n* = 795)	Category 2 High hs-CRP and Low CA125 (*n* = 786)	Category 3 Low hs-CRP and High CA125 (*n* = 1226)	Category 4 High hs-CRP and High CA125 (*n* = 1236)	*p*-Value
Age, years	75.8 ± 10.4	75.9 ± 10.3	73.5 ± 11.8	73.9 ± 11.2	<0.001
Male, *n* (%)	386 (48.6)	414 (52.7)	697 (56.9)	713 (57.7)	<0.001
**Medical history**					
Hypertension, *n* (%)	665 (83.6)	682 (86.8)	905 (73.8)	955 (77.3)	<0.001
Diabetes mellitus, *n* (%)	341 (42.9)	362 (46.1)	519 (42.3)	547 (44.3)	0.382
Dyslipidemia, *n* (%)	470 (59.1)	424 (53.9)	628 (51.2)	653 (52.8)	0.005
HFpEF, *n* (%)	215 (65.5)	217 (70.5)	134 (41.6)	182 (51.3)	<0.001
eGFR (MDRD formula), mL/min/1,73 m^2^	65.0 ± 26.7	60.4 ± 26.1	64.9 ± 30.7	59.8 ± 27.4	<0.001
Ischemic heart disease, *n* (%)	266 (33.5)	258 (32.8)	373 (30.4)	401 (32.4)	0.474
Peripheral edema, *n* (%)	406 (51.1)	424 (53.9)	832 (67.9)	866 (70.1)	<0.001
Pleural effusion, *n* (%)	253 (31.8)	259 (33)	730 (59.5)	741 (60)	<0.001
First hospital admission, *n* (%)	556 (69.9)	488 (62.1)	883 (72)	822 (66.5)	<0.001
NYHA III-IV, *n* (%)	106 (13.3)	137 (17.4)	187 (15.3)	232 (18.8)	0.006
Atrial fibrillation, *n* (%)	324 (40.8)	335 (42.6)	588 (48.0)	617 (49.9)	<0.001
**Vital signs**					
Heart rate, bpm	92.2 ± 26.9	94.9 ± 27.1	96.4 ± 27.6	98.5 ± 27.8	<0.001
SBP, mmHg	146.8 ± 31.4	146.1 ± 31.7	141.7 ± 29.9	141.1 ± 30.9	<0.001
DBP, mmHg	81.3 ± 20.3	79.3 ± 19.2	80.2 ± 17.9	79.0 ± 18.9	0.049
**Echocardiography**					
TAPSE	19.7 ± 3.6	19.5 ± 3.4	17.9 ± 3.6	18.0 ± 3.8	<0.001
Septum, mm	12.4 ± 3.3	12.1 ± 2.4	11.6 ± 2.6	11.8 ± 2.6	<0.001
DTDVI, mm	54.0 ± 9.2	53.1 ± 8.8	55.2 ± 9.5	54.7 ± 9.3	<0.001
LVPW, mm	11.6 ± 1.9	11.5 ± 1.9	11.1 ± 2.1	11.3 ± 2.0	<0.001
**Laboratory**					
Hemoglobin, g/dL ^a^	12.9 (11.6–14.1)	12.3 (11.0–13.7)	12.6 (11.3–13.9)	12.0 (10.8–13.6)	<0.001
Creatinine, mg/dL ^a^	1.1 (0.8–1.4)	1.1 (0.9–1.5)	1.1 (0.9–1.4)	1.2 (0.9–1.6)	<0.001
Sodium, mEq/L	138.8 ± 3.9	138.4 ± 4.4	138.8 ± 4.6	137.5 ± 4.8	<0.001
NT-proBNP, pg/mL ^a^	2541.4 (1292–4583)	3254.6 (1791–6000)	3889.8 (2093–7843)	5283.8 (2765.8–10,566)	<0.001
hs-CRP, mg/L	6.0 ± 3.8	58.7 ± 52.9	6.8 ± 4	53.9 ± 52	<0.001
CA125, U/mL ^a^	18.1 (11.6–25.9)	18.7 (12.7–26)	96.5 (58.8–174)	95.1 (56.8–159)	<0.001
**Treatment**					
ACE-I/ARB/ARNI, *n* (%)	505 (63.5)	458 (58.3)	756 (61.7)	704 (57)	0.010
MRA, *n* (%)	259 (32.6)	265 (33.7)	417 (34.0)	362 (29.3)	0.057
Diuretics prescribed at discharge, *n* (%)	719 (90.9)	724 (93.3)	1162 (95.3)	1149 (93.8)	0.001
Beta-blockers prescribed at discharge, *n* (%)	568 (71.4)	513 (65.3)	886 (72.3)	849 (68.7)	0.005
Statins prescribed at discharge, *n* (%)	465 (58.5)	389 (49.5)	606 (49.4)	570 (46.1)	<0.001

^a^ Value expressed as means ± standard deviations, medians (percentile 25–percentile 75,) or numbers (percentages). ACE-I: Angiotensin converting enzyme inhibitors; ARB: Angiotensin receptor blocker; ARNI: Angiotensin receptor–neprilysin inhibitor; CA125: Antigen carbohydrate 125; DBP: Diastolic blood pressure; DTDVI: Left ventricle telediastolic diameter; eGFR: Epidermal growth factor receptor; hs-CRP: High-sensitivity C-reactive protein; HFpEF: Heart failure with preserved ejection fraction; LVPW: Left ventricular posterior wall; NT-proBNP: N-terminal pro-B-type natriuretic peptide; MRA: Aldosterone receptor antagonists; NYHA: New York Heart Association; SBP: Systolic blood pressure; TAPSE: Tricuspid annular plane systolic excursion.

## Data Availability

The datasets presented in this article are not readily available because they are part of an ongoing study. Requests to access the datasets should be directed to the corresponding author.

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
