# Peer review of "Modulating Effect of Carbohydrate Antigen 125 on the Prognostic Value of High-Sensitivity C-Reactive Protein in Heart Failure"

_biomolecules, 2025, doi:10.3390/biom15091260_

Round 1
Reviewer 1 Report
Comments and Suggestions for Authors
The manuscript is well-organized and well-written. Furthermore, conclusions are inferred by data collected from a large cohort of patients.
The only doubt I have concerns the hsCRP threshold used by the authors. In the literature different levels are reported, but usually all < 10 mg/L. The authors should probably discuss the reson of their choice and, in the case, if the selection of more restrictive concentrations would have modified the overall output.
Reviewer 2 Report
Comments and Suggestions for Authors
In this manuscript, “ Modulating effect of carbohydrate antigen 125 on the prognostic value of high-sensitivity C-reactive protein in heart failure” by Santas et al. reported the interaction between CA125 and hsCRP in predicting adverse outcomes in acute heart failure (AHF). The study addresses an important and clinically relevant question: whether dual biomarker integration improves prognostic accuracy compared to single biomarkers. The work is supported by a large, multi-center cohort and robust statistical analyses. Although, authors had demonstrated some preliminary, the in vivo experiments were missing from this manuscript. Therefore, I would suggest authors might take at least a major revision before publication. Here are the comments and suggestions:
- In this study, CA125 and hsCRP were measured only once within 24 hours of admission, making it impossible to capture the trends of both biomarkers throughout the treatment course.WHY?
- The CA125 cut-off (35 U/mL) was derived from previous literature, but cross-platform harmonization and population applicability have not yet been validated.
- Further explanation is needed on how CA125 biologically amplifies the association between hsCRP and clinical outcomes, and the significance of such interaction for interpreting sensing signals.
- Tables 1 and 2: Move less critical variables to a supplementary appendix to improve clarity and focus in the main manuscript, while keeping key parameters in the primary tables.
Reviewer 3 Report
Comments and Suggestions for Authors
I recommend:
1. to use the same kind of shorten form of word for "high sensitivity C reactive protein" both in text (you use "hsCRP") as well as in tables (you use "hs-CRP");
2. to type in the same manner (use an interval bilaterally) in all cases with numbers (it is missed in line 34, 37, 116, 117, 118, 119, 123, 146, 152, 155 etc.) throughout the whole text.
Based hypothesis, interesting results and rational conclusions, correctly presented limitations, well done manuscript.
I suggest you proceed in future with deeper investigation of your hypothesis (look "Limitations").
Round 2
Reviewer 1 Report
Comments and Suggestions for Authors
no further comments